# Evaluation of Maternal Genetic Background of Two Hungarian Autochthonous Sheep Breeds Coming from Different Geographical Directions

**DOI:** 10.3390/ani12030218

**Published:** 2022-01-18

**Authors:** András Gáspárdy, Petra Zenke, Endre Kovács, Kata Annus, János Posta, László Sáfár, Ákos Maróti-Agóts

**Affiliations:** 1Department for Animal Breeding and Genetics, University of Veterinary Medicine, István u. 2, H-1078 Budapest, Hungary; kovacsdr.endre@gmail.com (E.K.); maroti-agots.akos@univet.hu (Á.M.-A.); 2Rex Pet Clinic, Lakkozó u. 13, H-1048 Budapest, Hungary; kata.annus@gmail.com; 3Department of Animal Husbandry, University of Debrecen, Böszörményi u. 138, H-4032 Debrecen, Hungary; postaj@agr.unideb.hu; 4Hungarian Sheep- and Goat Breeders’ Association, Lőportár u. 16, H-1134 Budapest, Hungary; safarlaszlo@majusz.hu

**Keywords:** maternal lineages, founder sampling method, cytochrome b gene, control region, Tsigai, Cikta

## Abstract

**Simple Summary:**

By the 19th century, adequately producing, independent domestic animal breeds had developed in many regions around the globe. However, from the middle of the 20th century they have largely been replaced by high-performing, specialized, single-purpose cosmopolitan breeds. Breed maintenance is an activity aimed at rescuing old breeds from the threat of extinction. This process includes recording the valuable traits of a rare breed, specific diversity conservation selection, and utilization in the original production environment. Additionally, it deals with the history of breeds and the study of their genetic makeup. The aim of this paper is to evaluate the maternal genetic background of two autochthonous sheep breeds in Hungary.

**Abstract:**

The aim of our research was the evaluation of the maternal genetic background of two Hungarian autochthonous sheep breeds of different geographical origin. A major argument for the preservation of endangered animal breeds is their documented past and historical importance. These also include the registration of pedigree data. This is the first study to evaluate and compare Tsigai and Cikta sheep in Hungary. Our investigation is based on two complete sequences of mitochondrial DNA (cytochrome b gene and control region). Our research was performed on these two sheep breeds with markedly different breed histories and breed characteristics to determine a possible common maternal genetic background, as ultimately the origin of both breeds can be traced back to Asia Minor. Between 2015 and 2017, a total of 203 biological samples were taken using a newly introduced founder sampling method. We found that the prevailing haplogroup B accounted for over 80% of both breeds, strengthening the common ancestral root. However, the pairwise genetic differentiation estimates (K_ST_) calculated using the sequence-based statistics for cytochrome b gene and control region were 0.034 and 0.021, respectively (both at level *p* < 0.05); thus, revealing genetic differentiation in both sequences between the Tsigai and Cikta. We note that the known different history of the breeds is clearly justified by the currently studied deviations in their maternal genetic background.

## 1. Introduction

Throughout the history of Hungarian sheep breeding, the initial primitive types were slowly replaced by new breeds. To the best of our knowledge, the development of these breeds was due to the following factors: firstly, the improvement of the old local sheep populations and secondly, the spread of foreign sheep in Hungary with the additional possibility of mixing among these. By the end of the 18th century, these breeds had become the native, autochthonous or heritage breeds of the country. Subsidies have been granted in an effort to preserve these undemanding, multipurpose fallow breeds that have been an endangered genetic resources since the middle of the 20th century. The following breeds make up our heritage breeds: Racka (in black and in white color variants), Gyimesi Racka (or Turcana, in the Romanian language Ţurcană), Cikta, Tsigai and Milking Tsigai. In this paper, we explore and compare the history and characteristics of the Tsigai [1] and Cikta [2] breeds (see Appendix A)

Mitochondrial DNA (mtDNA) is a special hereditary substance because it is not found in the nuclear chromosome within the cell nucleus, but it is a constituent of the chromosomes of mitochondria found in the cytosol. It is also special because it is circular in shape and does not recombine. In sheep, the coding region of 37 genes is located in the mitogenome; these include two ribosomal RNAs, 22 transfer RNAs, and 13 mitochondrial proteins. Another interesting factor is that some coding regions overlap. The mitogenome consists of approximately 16,500 base pairs. Exploring rare mutations that cause mitochondrial genetic disorders (e.g., neurodegenerative diseases, cancer, and diabetes [3]) is a major area of mitogenomic research today.

The unfolding of the microevolutionary web of our domesticated animals and the exploration of the origins of the developed breeds (e.g., [4]) is also worth mentioning.

The cytochrome b gene codes a protein that constitutes one of the 11 components of groups of proteins called complex III. The complex III is involved in the part of the oxidative photophosphorylation process in which the main energy source of the cell, adenosine triphosphate (ATP), is formed [5].

Pal et al. [6] were the first to report the association of the cytochrome b (Cyt b, CYTB) gene with disease traits in sheep. Mutations of Cyt b gene interfere with the site of heme-binding and calcium-binding domain, which is essential for the electron transport chain, resulting in anemia and malfunction of vital organs. This discovery raises the possibility that the sheep may be used as a model of humans.

It was observed that the cytochrome b gene is mutable in comparison with other mtDNA coding regions. This diversity of the CYTB gene is used in taxonomic studies to demonstrate the differences between species being compared. It is one of the most advantageous genes in mammals because its sequence is well known [7]. Using these comparative data, both bloodlines and family relationships can be determined.

The mitogenome also includes a short non-coding sequence, the control region (CR) [8]. The control region plays a role in initiating transcription and translation [9]. The hypervariable sites of the control region mutate significantly more frequently. Therefore, the CR is the subject of phylogenetic research, such as testing of species and populations based on maternal relatedness (e.g., [10]), forensic genetics (e.g., [11]) and studies of human races (e.g., [12]).

Based on the sheep mtDNA control region, researchers have so far identified seven haplogroups, five of which (A, B, C, D, E) still exist today, and two of which (F and G) are extinct [13].

Haplogroup C was separated from haplogroups A and B approximately 0.42–0.76 million years ago (Mya) based on CR analysis and about 0.45–0.75 million years ago (Mya) considering the CYTB sequences [14]. If solely hypervariable CR is taken into account in dating estimation, the separation time of sheep haplogroups was earlier [15]. However, the accuracy of the estimate may be uncertain due to the high incidence of recurrent mutations (e.g., [16]). Research on human mtDNA suggests that the exclusive consideration of CR may bring misleading results [17].

Haplogroup A is typical of Asian sheep, while haplogroup B is more common in European sheep. Haplogroup C shows a wide geographical distribution on these two continents (e.g., [18,19]).

Haplogroups D and E are the rarest groups and have only been identified in recent fossil and living samples from Turkey, the Caucasus [10,20,21], Europe [22,23], Tibet [24], and Iran [25]. Bone remnants of 4000–1000-year-old sheep in the Altai were studied by Dymova et al. [13], who found all five CR haplogroups that still exist today. Because of this diversity, it has been concluded that the Altai region has been a migratory area for many sheep and peoples in the past.

The combined use of CR and CYTB may meet the requirements for characterizing and comparing the haplogroup and haplotype of sheep breeds. Thus, this information also gains space in the research of the genetic structure of a breed (subspecies, variety, family). Examining the haplotypes (mutations, nucleotide diversity, genetic differentiation), alongside establishing the origin of the compared breeds [26], may shed light on the expansion of the breeds (e.g., [27,28]).

The two breeds we wish to investigate differ significantly in their external characteristics, their economically important traits, and their history. Taking into account the known differences between them, our aim is to map and compare the maternal genetic background of the two breeds originating from the same ancestral root, although developed at different geographical sites. We hypothesize that we will not find a considerable difference based on the conservative and coding cytochrome b gene sequence, however, according to the much more mutable, non-coding control region sequence there will be comparable differences between the Tsigai and Cikta coming to Hungary from the east and west, respectively.

## 2. Materials and Methods

### 2.1. Sampling

Full pedigrees provided information to achieve the so-called “founder sampling” method during the collection of biological samples. In this case, sample collection was preceded by processing the entire pedigree for both Tsigai [29] and Cikta [30], using the flock-books, which were re-established in 1995 and 2000, respectively. For these breeds, we obtained samples from the live female representatives of the oldest families (with 6-7-8 ancestral generations in Tsigai and with 4-5-6 ancestral generations in Cikta). The proportions of the sampled individuals in Tsigai and Cikta accounted for 9.3% (*n* = 134) and 20.5% (*n* = 69) of the seed stock of the breeds, respectively.

The live member of the families that provided the sample was selected so that as few farms as possible had to be visited for sampling in 2015 and 2017. Thus, the biological samples were obtained in nine Tsigai and three Cikta populations (see Appendix A). The blood samples (taken from vena jugularis) of the chosen individuals came from the samples annually taken in the course of routine veterinary procedures to detect any infectious disease (*Ovine brucellosis*). The further use of samples obtained during clinical veterinary procedures for research purposes is not considered an animal experiment under Law No. XXVIII of 1998 and Government Decree No. 40/2013, so no ethical permit is required.

Blood samples were frozen in tubes containing EDTA as soon as possible after collection. They were stored at −20 °C in a freezer until DNA was extracted.

### 2.2. MtDNA Processing

DNA was isolated using the GenElute Blood Genomic DNA Kit (Sigma-Aldrich, St. Louis, MO, USA) according to the manufacturer’s instructions. A 1.5% agarose gel mixed with GelRedTM Nucleic Acid Gel Stain (Biotium, Fremont, CA, USA) was used for quality and for a semi-quantitative measurement of the extracted DNA. Purified DNA was stored at 4 °C until subsequent analysis.

The 25 μL PCR reaction mixture prepared for each sample contained 5 μL DreamTaq™ Green PCR Master Mix (ThermoFisher Scientific, Waltham, MA, USA), 1-1 μL forward and reverse primers (10 μM), 1 μL BSA (20 mg/mL), and approximately 10 ng DNA template and PCR grade water to volume.

For both the cytochrome b gene and control region, primers [20,23,31] were selected to evaluate the entire sequence (1140 and 1180 base pairs, respectively; see Appendix A).

To amplify the DNA sequences, a programmable Thermal Cycler 2720 PCR equipment (Applied Biosystem, Waltham, MA, USA) was chosen. For the purification of PCR products a SIGMA GenE-luteTM PCR Clean Up Kit (Sigma-Aldrich, St. Louis, MO, USA) was used according to the protocol.

For the sequencing reaction, a BigDye^®^ Terminator v.3.1 Cycle Sequencing Kit (ThermoFisher Scientific, Waltham, MA, USA) was applied in the manner recommended by the manufacturer. An ABI Prism 3130XL Genetic Analyzer (Applied Biosystems, ThermoFisher Scientific, Waltham, MA, USA) was applied for sequence detection, according to the manufacturer’s guidelines. Sequence data were analysed using Sequencing Analysis Software 5.1 (Applied Biosystems, ThermoFisher Scientific, Waltham, MA, USA) and aligned by SequencherTM 4.1.2 software (Gene Codes Corp, Ann Arbor, MI, USA).

### 2.3. Statistical Analysis

First, we used DNAsp v6.0 software (Rozas et al. & University of Barcelona, Barcelona, Spain) [32] to determine the number of polymorphic sites in the entire study sample and in the breeds, calculated the mean nucleotide difference within the groups and compared the two breeds based on pairwise genetic differentiation estimates (K_ST_). Then, we used the test developed by Fu and Li [33] to evaluate mutations in CYTB and CR sequences, and for the nucleotide diversity of CYTB and CR, we used the Tajima D test developed by Tajima [34] to clarify the genetic equilibrium status of the population.

The resulting sequences were aligned with MEGA-X (MEGA, Tokyo Metropolitan University, Hachioji, Japan) [35]. The number of corrected base substitutions within the sequences was determined by the method of Jukes and Cantor [36,37].

The species distribution of CR haplotypes was plotted using Network 10.2.00 software (fluxus-engineering.com, accessed on 31 May 2020); [38]). The selection of samples into haplogroups was performed by comparing our obtained sequences with the reference sequences of the gene bank.

All novel sequences of cytochrome b gene and control region used in this study were deposited in the GenBank public database under following accession numbers: OK572693-OK572826 (CYTB in Tsigai, *n* = 134), OK572625-OK572692 (CYTB in Cikta, *n* = 68), OK572827-OK572956 (CR in Tsigai, *n* = 130) and MW427980-MW427983, MW427991, MW427993-MW428057 (CR in Cikta, *n* = 69).

## 3. Results

For CYTB, the number of nucleotide sites and number of sequences were 1140 and 199, respectively. The number of mutations was 50 (13 singleton with positions: 110, 397, 435, 459, 642, 650, 699, 717, 762, 871, 933, 945, 1065 and 37 parsimony ones).

For CR, the number of nucleotide sites and number of sequences were 1180 and 199, respectively. The number of total mutations was 185 (25 singleton with positions: 8, 58, 160, 161, 174, 265, 283, 309, 315, 332, 335, 359, 379, 483, 538, 544, 581, 691, 710, 973, 996, 1075, 1132, 1144, 1155 and 160 parsimonies, not listed here).

The number of individuals sampled and polymorphic positions per breeds are shown in Appendix A. There were 6 shared mutations, 12% only for CYTB and 85 shared mutations, 46% for CR.

The total number of haplotypes in the whole study population was 40 for CYTB and 94 for CR. Of the CYTB haplotypes, 33 were distinguishable in the Tsigai and 11 in the Cikta. Of the CR haplotypes, 66 occurred in Tsigai and 29 in Cikta.

A comparison of the two breeds is presented in Table 1 for haplotype diversity (Hd), the average number of nucleotide differences (k) and average nucleotide diversity (π). It can be seen that the Cikta is characterized by a greater diversity than the Tsigai for both sequences (CYTB and CR) because of the higher average number of nucleotide differences (3.384 and 21.267, respectively) and greater nucleotide diversity with Jukes and Cantor (2.98 × 10^−3^, and 18.56 × 10^−3^, respectively). Only the haplotype diversity was found to be the same for both sequences (0.839 and 0.973 in Tsigai, 0.858 and 0.961 in Cikta).

The average number of pairwise nucleotide differences between the two populations for CYTB and CR was 2.422 and 18.466, respectively.

The pairwise genetic differentiation estimates (K_ST_) calculated using the sequence-based statistic for CYTB and CR were 0.034 and 0.021, respectively (both at level *p* < 0.05); thus, the genetic differentiation in both sequences between the Tsigai and Cikta was found.

The corrected number of base substitutions (Dxy) was, for CYTB and CR, 2.46 × 10^−3^ and 15.81 × 10^−3^, respectively.

Figure 1 shows the mismatches between pairwise combinations of mtDNA CR haplotypes according to the breeds. The dotted lines represent the expected distribution under expansion, the gray (Tsigai) and orange (Cikta) dots represent the observed distributions under expansion. The observed distribution curve seems to be almost unimodal in Tsigai, which indicates that it might have undergone a recent demographic expansion. A skewed unimodal distribution is generally associated with a recent sudden expansion. In Cikta, for the bimodal distribution, the separate peaks show that two dominating groups of haplotypes exist in that breed, which is thought to be related to a more constant population size.

The number of CR haplogroups identified was three. The classification of individuals into CR haplogroups was determined in the following proportions. The most populous of the haplogroups was B (of which 126 are Tsigai and 56 are Cikta), followed by A, with 4, and 8 individuals, respectively. Furthermore, 6 Cikta individuals were included in haplogroup C.

The network of the connections between the CR haplotypes and the reference CR haplogroups computed by the median-joining method illustrates well that the haplotypes are definitely located around the reference haplogroup B (Figure 2).

The Tajima D-test value for CYTB was −2.124, statistically slightly significant (*p* < 0.05) and for CR it was −1.480, which was statistically non-significant (*p* > 0.10).

For CYTB, the Fu and Li’s D* and F* tests also resulted in non-significant values of 0.582 (*p* > 0.10) and 0.121 (*p* > 0.10), respectively, in the whole sample set investigated. The Fu and Li’s D* and F* test values for CR were as follows: 0.620 (*p* > 0.10) and −0.468 (*p* > 0.10).

For both CYTB and CR, the Fu’s Fs statistic resulted in significant negative values of −33.177 (*p* < 0.001) and −41.261 (*p* < 0.001), respectively.

## 4. Discussion

For the CR, in the evaluation of Kusza et al. [40], the average nucleotide diversity of two heritage breeds in the Carpathian Basin, Gyimesi Racka (bred in Hungary) and Turcana (bred in Romania) was lower (both 5 × 10^−3^, which can be explained also by the lower number of bp used) than that of all breeds currently studied. In this study, the Tajima D-values of both breeds were statistically proven to be in the minus range (Gyimesi Racka −1.734 and Turcana −1.814).

For Hd, Dudu et al. [41] found a 0.958 variant of the Hungarian Racka bred in Romania, taking into account the CR and CYTB gene together. In this research, not just the more frequently mutated control region, but also the longer cytochrome b sequence, may have contributed to the relatively high value of that single breed variant. In the analysis conducted by Pariset et al. [42], the average number of nucleotide differences and corrected number of base substitutions (Dxy) among the Albanian breeds (Bardhoka, Ruda, Shkordane) for CR varied from 7.647 to 10.567 (lower than our), and from 18 × 10^−3^ to 26 × 10^−3^ (higher than our), respectively. Furthermore, they observed a high common level for nucleotide diversity (π) of 21 × 10^−3^ in the three investigated breeds (with a total of 53 individuals), which is higher than that of Tsigai and Cikta in our work. In the Ruda breed (which is a relative of Tsigai), they found no haplogroup C, while in the other two they detected around 15%. In the study of Niemi et al. [43], haplotype diversity (Hd = 0.94) and nucleotide diversity (π = 15.51 × 10^−3^) in the Serbian Pramenka breed had similar values to ours; however, the average number of nucleotide differences (k = 8.11) was significantly lower.

In our processing of the genetic pool of the two breeds, Cikta proved to be more diverse with regard to the average number of mutations per individual, average number of nucleotide differences, and the average nucleotide diversity (JC). We can conclude that its origin is also more complex. The two peaks that appeared in the mismatch distribution of the Cikta further reinforce the more diverse maternal background of this breed. According to Alvarado-Bremer et al. [44], the bimodal peaks suggest the occurrence of two distinct sub-populations, which brings into question the interpretation of a breed as a single genetic unit. The first mode would represent recent intra-clade pairwise differences, whereas the second one would likely represent more ancient inter-clade pairwise differences. If this is the case, then the population of today’s Hungarian Cikta consists mostly of individuals with minor intra-breed genetic differences, and to a lesser, but remarkable extent, of individuals with greater genetic difference. In contrast, the population of the Tsigai breed is almost without exception composed of slightly different individuals.

Furthermore, the Tajima D-test and Fu’s Fs statistic performed on the entire study population confirmed a significant segregation of the cytochrome b gene sequence. Contrary to our expectation, we found a non-significant (Tajima D-test) differentiation in the control region. A significant positive value would be an indicator of the loss of genetic diversity in a given population (bottleneck effect) or degradation to subpopulations. Meanwhile, a significant negative value signifies an excess of low frequency polymorphisms relative to expectation or population-size expansion [45]. A non-significant D value reflects a stable population.

However, the more sensitive Fu’s Fs statistic confirmed separations according to both sequences. The significant negative values of Fu’s Fs statistic (CYTB −33.177 and *p* < 0.001, CR −41.261 and *p* < 0.001) demonstrate foreign gene flows after a spatial expansion.

The value of Fu’s Fs statistic can be considered as medium in our study, since in the clusters of other similar sheep research, it decreased by a smaller (e.g., −7.48, *p* = 0.001 [24]; −10.88, *p* < 0.001 [18]) or a greater extent (−76.28, *p* < 0.001 [46]) within the negative range, with a similarly low probability of error. The results of the test support the conclusion that genetic segregation under co-processing also means the separation of the Tsigai and the Cikta breeds from each other.

In our investigation, we distinguished three haplogroups. The haplogroup B (typical for European sheep domesticated in Near East) prevailed in both breeds (Tsigai and Cikta) by 97% and 81%, respectively. The Copper Age sheep in Europe were also found to belong to haplogroup B, as a link between peat sheep and its distant ancestor in Near East was found by Oliveri et al. [47].

In our case, the haplogroup A appeared with a minor frequency. The Cikta sheep is distinguished by a proportion of haplogroup A (12%), while this haplotype is also present in the Tsigai, but much less frequently (3%).

The current occurrence of haplogroup A in the Carpathian Basin cannot be considered unexpected. The territory of Hungary, which is probably one of the earliest continental sites for the production of fine wool, provides the oldest sheep remains in Europe belonging to the haplogroup A. The Bronze Age sheep remains found by Sabatini et al. [48] in Hungary (Százhalombatta-Földvár, 1500 BC) that were classified into haplogroup A, may serve as evidence that a new breed of sheep was introduced to Europe in the Bronze Age to improve the productivity of local flocks. The haplogroup A descends from the Asiatic mouflon (*Ovis orientalis*) and it is mainly present in Asian breeds, thus, it is, characteristic of the breeds in the Indian subcontinent (77%; [49]). Ewes of haplogroup A which entered into Carpathian Basin during this time may have served as a partial background for today’s Hungarian indigenous breeds.

During sheep phylogenetic development, haplogroup C was first separated from B and A. In our study, we observed the occurrence of haplogroup C only in the Cikta breed. The 9% occurrence of haplogroup C also supports the view that the Cikta has a more complex maternal background than the Tsigai. This means that ewes may have entered Hungary in the past from the central areas of Asia, from the Caspian See region [49], Mongolia [50]) and China [14,19,46] wherein the haplogroup C is characteristic.

The wild sheep found no suitable habitat and did not survive due to climatic and environmental conditions in Europe by the end of the Stone Age. The European spread of domestic sheep could have occurred through several highly probable routes, partly along the Danube Valley, via the Carpathian Basin and along the Eastern European plain [51]. The domesticated sheep of haplogroup B arrived in Europe with Indo-Germanic tribes from Anatolia and populated this continent. At a similar time (around 6500 BC), during the expansion of the Altai ethnic groups, shepherds and their sheep reached the southern Carpathians, too, which made it possible for haplotype C to enter Europe. Later, as a consequence of a significant steppe effect, haplogroup A sheep were introduced to Europe in the Bronze Age with the population of the Jamna culture. During the Iron Age in the Carpathian Basin (560 BC), Scythian peoples came to the territory of present-day Hungary, who were more likely to bring sheep from the east, which sheep could also be classified as haplogroup C. Regarding the autochthonous sheep of Hungary, the age of the Turkish occupation (1552–1693) can be the last “weakly documented” migration period. This was the time when fat-tailed sheep that also show haplogroup C, such as Karagül and Kivircik [21] may have merged into local populations, that is, the haplogroup C was transferred via Anatolian mediation. So, it is likely that the haplogroup C can be detected in other native sheep varieties in Hungary or in other surrounding countries. Studies [52,53] conducted in neighboring countries dealt with the classification of domestic sheep breeds into haplogroups but none revealed the occurrence of the haplogroup C until now.

The haplogroup C had dispersed from the inner core of Asia not only to central Europe. According to the literature, haplotype C can also be found in Caucasus [10] and Georgia [54] as well as in India [55], Asia Minor [20], Northern Anatolia [56]), in the Mediterranean region of Europe [42,57,58] and Africa [59,60,61]. A third route of the European spread of sheep was a direct one from Asia Minor via the Mediterranean Sea.

We have not detected clear haplogroups E and D in our current investigation.

The latest common gene pool of the two breeds can be traced back to Anatolia. We can assume that the diversification of the common gene pool took place 7500 years ago. After all, the early ancestors of the Cikta entered the central part of the European continent from the Balkans, and from here they reached the Carpathian Basin in the early 18th century. The ancestors of the Tsigai made their way from Anatolia directly to the Balkans and the Carpathian Basin over the last few centuries. Now, we conclude that the demographic history of the two breeds differs from each other, a clear sign of which is the significantly differing maternal background. Obviously, the migration of the two populations could not have been entirely independent from various genetic effects, because the genetic background of both breeds was not only modified by the occurrence of natural mutations but also by gene flow from other sheep herds depending on space and time.

For the success of mitochondrial genome examinations, prior knowledge of pedigree data is essential for the identification of maternal lineages and for organized sampling. We consider the founder sampling method, which can be used to calculate the true values of mitochondrial diversity, to be a form of truly representative sampling. Furthermore, finding unique haplotypes of rare families is most likely only feasible in this way. The question arises, if a breed is considered to have a uniform genetic background, whether individuals with a haplotype that differs from their breed-mates should be excluded from preservation of native breeds, or if they are worth maintaining.

Furthermore, by identifying families, our work also brings attention to the importance of the maintenance of maternal lineages and within-family selection. Increased consideration of the maternal side is also supported by the fact that the proportion of ewes and rams in breeding herds is in favor of females. Additionally, ewes stay in breeding for longer than rams. Thus, they may be better custodians of the display and transmission of genetic diversity.

## 5. Conclusions

In the course of our research, the fact that complete sequences were analyzed in both the cytochrome b and control regions certainly provided more reliable results than the use of only one sequence or a partial sequence. In addition to these reliable results, the number of individuals in the breeds was sufficiently large considering the sample size in other similar studies referred in our paper. Furthermore, due to the introduction of the founder-sampling method as a basis for sampling, the maternal background could be characterized more realistically than by random sampling.

We were faced with the fact that our working hypothesis, which was based on the breed histories and experiences with mtDNA sequences, was only partially fulfilled. Contrary to expectations, we showed a statistically significant genetic separation between Cikta and Tsigai not only in the control region but also in the cytochrome b gene. Based on the rare allele’s excess of CYTB and CR, both breeds demonstrate foreign gene flow after a spatial expansion. In addition to the European haplogroup B, the presence of haplogroup A was confirmed in both breeds, but haplogroup C only in the Cikta breed.

## Figures and Tables

**Figure 1 animals-12-00218-f001:**
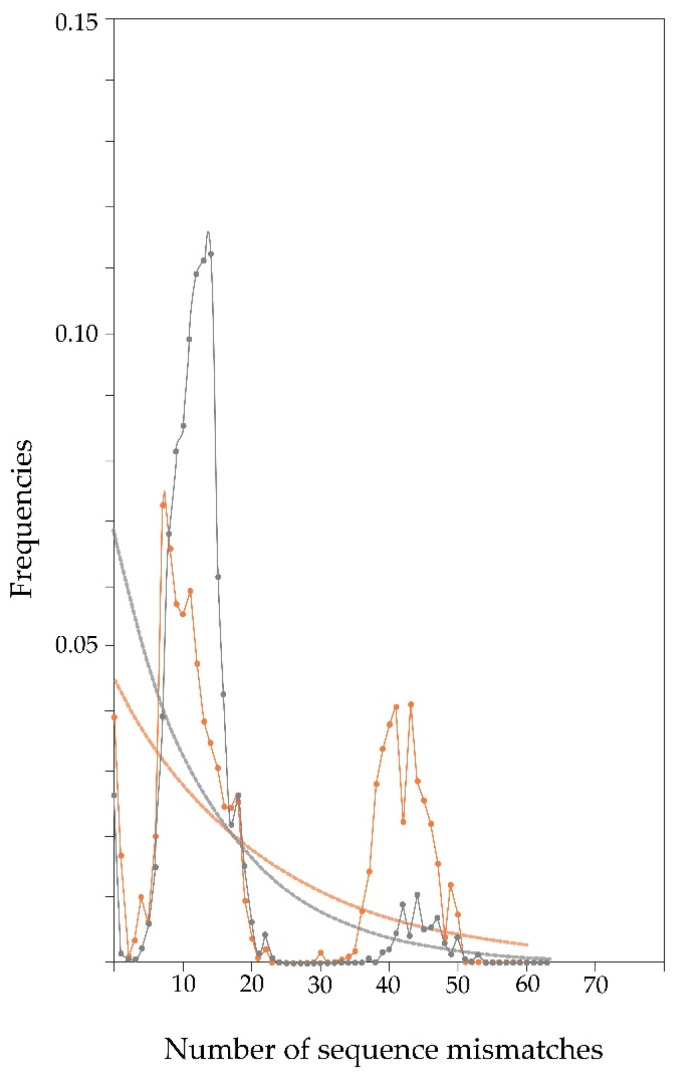
Frequency distribution of the number of sequence mismatches between pairwise combinations of Tsigai (gray) and Cikta (orange) mtDNA CR haplotypes.

**Figure 2 animals-12-00218-f002:**
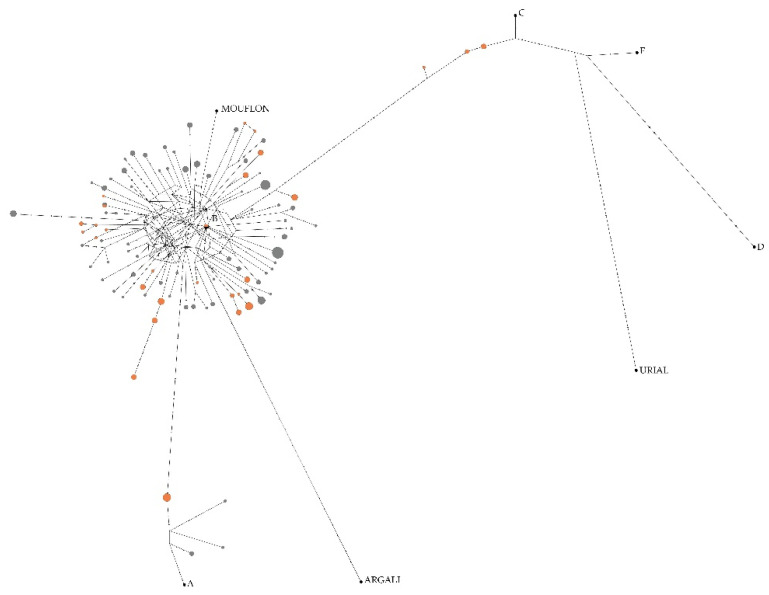
Connections between sample CR haplotypes and reference CR haplogroups by median-joining network. Legend: Tsigai (gray), Cikta (orange), different black colored spots with denomination are reference samples (A-HM236174, B-HM236176, C-HM236178, D-HM236180, E-HM236182 [17]; *O. musimon* Mouflon HM236184, *O. ammon* Argali HM236188, *O. vignei* Urial HM236186 [39]). The size of the circles is in proportion to the number of individuals observed per haplotype, and the lines crossed in the sections connecting the haplotypes indicate the number of mutations between them.

**Table 1 animals-12-00218-t001:** Values of haplotype diversity, k and π (JC) according to breeds for CYTB and CR.

Parameter	Tsigai	Cikta	Total
Haplotype diversity, Hd			
CYTB	0.839	0.858	0.866
CR	0.973	0.961	0.984
Average number of nucleotide differences, k			
CYTB	1.782	3.384	2.422
CR	13.675	21.267	16.760
Nucleotide diversity (Jukes and Cantor), π(JC)			
CYTB	1.57 × 10^−3^	2.98 × 10^−3^	2.13 × 10^−3^
CR	11.84 × 10^−3^	18.56 × 10^−3^	14.57 × 10^−3^

## Data Availability

The mtDNA sequence data are available from the NCBI GenBank repository.

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
