# Peer review of "Evaluation of Maternal Genetic Background of Two Hungarian Autochthonous Sheep Breeds Coming from Different Geographical Directions"

_animals, 2022, doi:10.3390/ani12030218_

Round 1

Reviewer 1 Report

I supported publication of this manuscript during the original submission. 
The authors have made improvements and the manuscript is now in a better condition.
However, the language problems persist and a detailed revision of the entire manuscript needs to be done to rectify this problem before final acceptance.

Author Response

Response to Rewiever 1:

“I supported publication of this manuscript during the original submission.

The authors have made improvements and the manuscript is now in a better condition.

However, the language problems persist and a detailed revision of the entire manuscript needs to be done to rectify this problem before final acceptance.”

  • Thanks to Reviewer 1 for appreciating the development of the manuscript compared to previous version. The manuscript was once again revised in detail.

Reviewer 2 Report

The manuscript has significantly improved in terms of methodology and description of results and authors' efforts are appreciated.  A few remaining concern are give as follows:-

Line 22. A clear objective statement may please be provided in the start

Lines 23-30. Few more lines be added for methodology

Lines 34-36. A clear result and conclusion statement may please be provided

Line 37. Only 3-5 Keywords are enough.

Lines 41-48. Provide references for your statements.

Line 53-56. Provide references for your statements.

Line 63. Writing “(e.g. [6])” many times is not great. Please revise and try to write in the form of story.

Lines 67-70. Provide references for your statements.

Line 361. Why? “We were faced with the fact that our working hypothesis, which was based on the 361 breed histories and experiences with mtDNA sequences was only partially fulfilled”

The confusion still persists on the coherence of objectives and results and that requires more clarity in write up.  

Author Response

Response to Rewiever 2:

“The manuscript has significantly improved in terms of methodology and description of results and authors' efforts are appreciated.  A few remaining concern are give as follows:-“

  • Thanks to Reviewer 2 for appreciating the development of the manuscript especially in the methodological description and communication of the results compared to previous version.

“Line 22. A clear objective statement may please be provided in the start”

  • We have included an objective statement.

“Lines 23-30. Few more lines be added for methodology”

  • We have added more information to the methodology section.

“Lines 34-36. A clear result and conclusion statement may please be provided”

  • The result statement starts from line 32.

“Line 37. Only 3-5 Keywords are enough.”

  • According to the Guide for authors the maximum number of keywords is 6.

“Lines 41-48. Provide references for your statements.”

  • More references are provided in the Appendix A, according to the request of a previous reviewer.

“Line 53-56. Provide references for your statements.”

  • This part has been reconstructed according to the Editor’s wish.

“Line 63. Writing “(e.g. [6])” many times is not great. Please revise and try to write in the form of story.”

  • Where it was most reasonable, we listed the authors in form of story. The form a reference specifies that author names be replaces by numbers.

“Lines 67-70. Provide references for your statements.”

  • The requested reference has been included in the manuscript.

“Line 361. Why? “We were faced with the fact that our working hypothesis, which was based on the 361 breed histories and experiences with mtDNA sequences was only partially fulfilled””

  • This is clearly presented in this paragraph (chapter Conclusions).

“The confusion still persists on the coherence of objectives and results and that requires more clarity in write up.  “

  • The confusions marked by the Reviewer 2 clearly have been corrected in the manuscript.

Reviewer 3 Report

Although the topic that this manuscript deals with is interesting. This rather looks as if it was an unfinished paper. English is difficult to follow at some points and sentences are so long sometimes, that you lose their meaning. This manuscript is worth considering but, after considerable work is done. Hence, I would go for a second round of revision.

Title:

Appropriate

Simple summary

I am sorry but this simple summary looks more like a short summary of local breeds importance rather than focus on the topic that the paper deals with.

Abstract

This section is better. However, this should be written as a single paragraph. It is true that I did feel that authors should put a bigger effort to make the abstract look like a real short summary of the sections of the paper in an organized manner, which it does not do as it is now.

Keywords: Appropriate

Introduction is a mess. It feels as if the authors jumped from one topic to another, which breaks the flow of the manuscript. After reading it, I end up confused about what the authors really aimed to determine through their research.

Lab M&Ms seem appropriately described, however, details on the total census of the current population of the breeds investigated in this paper may help to figure out the magnitude of the representativity of the sample used.

 Statistics seem to be appropriate.

From my point of view discussion is the best part of the manuscript although I did not like the style in which the authors wrote it. Especially, discussion seems to be divided in two parts: comparison to literature and comment on results from present research. This information should be better process, and comments on results by the present authors must be supported or counteracted by previous literature. I feel a lack of effort overall in this paper. It is as if the paper had not gone through the last round of revision by authors prior to sending, which is upsetting because, endangered local breed must be given the opportunity that they deserve in science.

Conclusions.

More reliable results than…what?

More realistically…again than what?

Again, what is the take home message of this paper?  Conclusion section should extend no further that 5 o 6 lines, not to or three paragraphs. Please rewrite.

Author Response

Response to Rewiever 3:

“Although the topic that this manuscript deals with is interesting. This rather looks as if it was an unfinished paper. English is difficult to follow at some points and sentences are so long sometimes, that you lose their meaning. This manuscript is worth considering but, after considerable work is done. Hence, I would go for a second round of revision.”

  • Thank you for finding the manuscript worthwhile. The manuscript has been revised as suggested by Reviewer 3.

“Title: Appropriate”

“Simple summary

I am sorry but this simple summary looks more like a short summary of local breeds importance rather than focus on the topic that the paper deals with.”

  • Following the author’s guidelines and using published papers as an example (e.g. Animals 2021, 11(12), 3599; https://doi.org/10.3390/ani11123599), we prepared the simple summary for the lay audience.

“Abstract

This section is better. However, this should be written as a single paragraph. It is true that I did feel that authors should put a bigger effort to make the abstract look like a real short summary of the sections of the paper in an organized manner, which it does not do as it is now.”

  • We have rephrased and reconstructed the abstract.

“Keywords: Appropriate”

“Introduction is a mess. It feels as if the authors jumped from one topic to another, which breaks the flow of the manuscript. After reading it, I end up confused about what the authors really aimed to determine through their research.”

  • We have reconstructed the introduction.

“Lab M&Ms seem appropriately described, however, details on the total census of the current population of the breeds investigated in this paper may help to figure out the magnitude of the representativity of the sample used.”

  • We have added the representation of the breeds to this chapter.

“Statistics seem to be appropriate.”

“From my point of view discussion is the best part of the manuscript although I did not like the style in which the authors wrote it. Especially, discussion seems to be divided in two parts: comparison to literature and comment on results from present research. This information should be better process, and comments on results by the present authors must be supported or counteracted by previous literature. I feel a lack of effort overall in this paper. It is as if the paper had not gone through the last round of revision by authors prior to sending, which is upsetting because, endangered local breed must be given the opportunity that they deserve in science.”

  • We have modified the chapter Discussion.

“Conclusions.

More reliable results than…what?”

  • We have supplemented this sentence.

“More realistically…again than what?”

  • We have supplemented this sentence.

“Again, what is the take home message of this paper?  Conclusion section should extend no further that 5 o 6 lines, not to or three paragraphs. Please rewrite.”

  • The first two paragraphs referring the direct results have been left. The third paragraph has been moved to chapter Discussion.

Reviewer 4 Report

This is potentially an interesting paper, investigating the relatedness and genetic history of two Hungarian sheep breeds. However, the manuscript is difficult to read and it is not always clear what the Authors meaning is. I report some examples and suggestions for improvement, which are not covering all issues.

Abstract

It is not usual to write using the plural third person, I suggest the Authors change it into “This is the first study comparing…”, or “Our study is the first comparing…” instead than “The authors are the first to evaluate and compare the…”

Introduction

Line 41: It is not clear what the difference between the primitive types and the subsequent breeds is, were the individual from the different breeds registered in a studbook?

Line 44: I suggest changing spread into import, since the sheep were not spreading by themselves.

Line 54: I suggest changing into “outside the cell nucleus”

Line 57: It is not clear what “next to…” means in this context, was the meaning “such as…”?

Lines 61-63: I suggest reformulating this sentence to improve its clarity.

Lines 74-75: “This discovery raises the possibility that the sheep may come into play as a model of man.” I suggest changing into “This discovery raises the possibility that the sheep may be used as a model for humans”.

Lines 84-85: “This unstable segment gives the basis for dating estimation in many mammalian species.” This sentence is unclear, what are “dating estimations”?

Line 107-108: This sentence should be reformulated to improve its clarity.

Results

Figure 2: The figure and the legend are not very clear, it would be better to explain that the grey dots are Tsigai individuals instead than writing Tsigai-gray. In Figure 1, it was Tsigai (gray). It should also be explained why the dots have different sizes and what the length of the connection lines represent.

Lines 235-236: what do indicate the significant negative values of Fs statistics?

Line 332I suggest changing “It is conceivable” into “It is likely”.

Discussion

Please explain what “gene reserve breeds” are.

Line 266-267 Please explain what the “community of a breed” is.

Conclusion

Lines 371-372 “The latest common gene pool of the two breeds can be traced back to Anatolia. According to the an extreme idea, the separation of the common gene pool can be assumed to have taken place 7,500 years ago.” This phrase has typos and should be rewritten to improve its clarity. Why is this idea extreme?

Ethical statement

There should be some information about ethical approval; did the Authors apply for approval of blood sampling technique before conducting the study? If not, were the blood samples collected from other purposes? This should be stated clearly.

Author Response

Response to Rewiever 4:

“This is potentially an interesting paper, investigating the relatedness and genetic history of two Hungarian sheep breeds. However, the manuscript is difficult to read and it is not always clear what the Authors meaning is. I report some examples and suggestions for improvement, which are not covering all issues.”

  • Thanks to Reviewer 4 for their general appreciation of the manuscript. Corrected linguistic errors are corrected.

“Abstract

It is not usual to write using the plural third person, I suggest the Authors change it into “This is the first study comparing…”, or “Our study is the first comparing…” instead than “The authors are the first to evaluate and compare the…”

  • The wording of the abstract has been transformed into a plural first person.

“Introduction

Line 41: It is not clear what the difference between the primitive types and the subsequent breeds is, were the individual from the different breeds registered in a studbook?”

  • The term “primive type” (unimproved) is commonly known in animal husbandry. We mean this for animals with low production. The description of animals that have already become breeds in our case is presented in Appendix A within the scope limits.

“Line 44: I suggest changing spread into import, since the sheep were not spreading by themselves.”

  • We want to keep the word “spread”, because in the 1700s the Cikta sheep came to Hungary with the settling Swabian people population. The use of the word “import” does not fit into this context.

“Line 54: I suggest changing into “outside the cell nucleus””

  • We have accepted this suggestion.

“Line 57: It is not clear what “next to…” means in this context, was the meaning “such as…”?”

  • We have accepted this proposal.

“Lines 61-63: I suggest reformulating this sentence to improve its clarity.”

  • We have accepted this suggestion.

“Lines 74-75: “This discovery raises the possibility that the sheep may come into play as a model of man.” I suggest changing into “This discovery raises the possibility that the sheep may be used as a model for humans”.”

  • We have accepted this proposal.

“Lines 84-85: “This unstable segment gives the basis for dating estimation in many mammalian species.” This sentence is unclear, what are “dating estimations”?”

  • We have reworded this sentence.

“Line 107-108: This sentence should be reformulated to improve its clarity.”

  • We have reworded this sentence.

“Results

Figure 2: The figure and the legend are not very clear, it would be better to explain that the grey dots are Tsigai individuals instead than writing Tsigai-gray. In Figure 1, it was Tsigai (gray). It should also be explained why the dots have different sizes and what the length of the connection lines represent.”

  • The requested explanation was added to the legend of Figure 2.

“Lines 235-236: what do indicate the significant negative values of Fs statistics?”

  • The meaning of a significant negative values of Fs statistics is mentioned in the chapter Discussion.

“Line 332 I suggest changing “It is conceivable” into “It is likely”.”

  • We have accepted this suggestion.

“Discussion

Please explain what “gene reserve breeds” are.”

  • Instead of “gene reserve breed”, we use the more popular term “heritage breed”.

“Line 266-267 Please explain what the “community of a breed” is.”

  • The word "community" has been deleted in this sentence.

“Conclusion

Lines 371-372 “The latest common gene pool of the two breeds can be traced back to Anatolia. According to the an extreme idea, the separation of the common gene pool can be assumed to have taken place 7,500 years ago.” This phrase has typos and should be rewritten to improve its clarity. Why is this idea extreme?”

  • We have reworded this sentence.

“Ethical statement

There should be some information about ethical approval; did the Authors apply for approval of blood sampling technique before conducting the study? If not, were the blood samples collected from other purposes? This should be stated clearly.”

  • The conditions for collecting the blood sample were clarified in the chapter Materials and Methods.

Round 2

Reviewer 3 Report

The efforts of authors for implementing the suggestion made is worth mentioning. However, conclusions, from my point of view are still contaminated by some results. Please revise.

Author Response

Response to Reviewer 3:

„The efforts of authors for implementing the suggestion made is worth mentioning. However, conclusions, from my point of view are still contaminated by some results. Please revise.“

  • The section containing the detailed results has been shortened.

Reviewer 4 Report

I found the revised version of the manuscript improved. I still think that the English should be checked for clarity and some statements should be moderated. Here are some examples (not covering all issues):

Lines 91-92: “If hypervariable CR is also taken into account in dating estimation, the separation time of sheep haplogroups was earlier [15].”

I understand what you want to say but this is not clear enough for a scientific article. “was” is in this case not the corrected tense and the whole structure of the sentence is not grammatically correct.

Legend of Figure 2: “The size of the circles is in proportion to the number of individuals observed per haplotype, and the lines crossed in the sections connecting the haplotypes indicate the number of mutations between them.” What does it mean “the lines crossed in the section connecting…”? Why are some lines continuous and some are dashed or dotted? A Figure should be self-explanatory.

Line 322 “The wild sheep found no habitat for themselves”. I suggest changing into “suitable habitat” and cut “for themselves”

Lines 334-335 “Regarding the autochthonous sheep of Hungary, the last “weakly documented” migration period can be the age of the Turkish occupation (1552-1693).” I suggest rephrasing starting with “The age of Turkish occupation (1552-1693) might represent the last….”

Lines 376-377 The Authors write “In addition to these reliable results, the number of individuals in the breeds was sufficiently large.” I wonder what “sufficiently large” means in this context, is it in comparison to previous research or to some guidelines?

Author Response

Response to Rewiever 4:

„I found the revised version of the manuscript improved. I still think that the English should be checked for clarity and some statements should be moderated. Here are some examples (not covering all issues):

Lines 91-92: “If hypervariable CR is also taken into account in dating estimation, the separation time of sheep haplogroups was earlier [15].”

I understand what you want to say but this is not clear enough for a scientific article. “was” is in this case not the corrected tense and the whole structure of the sentence is not grammatically correct.“

  • This comment was taken into account.

„Legend of Figure 2: “The size of the circles is in proportion to the number of individuals observed per haplotype, and the lines crossed in the sections connecting the haplotypes indicate the number of mutations between them.” What does it mean “the lines crossed in the section connecting…”? Why are some lines continuous and some are dashed or dotted? A Figure should be self-explanatory.“

  • The supposed misrepresentation in Figure 2 is due to an IT error. The original image sent to the Editor shows the object in question well.

„Line 322 “The wild sheep found no habitat for themselves”. I suggest changing into “suitable habitat” and cut “for themselves”“

  • This suggestion was accepted.

„Lines 334-335 “Regarding the autochthonous sheep of Hungary, the last “weakly documented” migration period can be the age of the Turkish occupation (1552-1693).” I suggest rephrasing starting with “The age of Turkish occupation (1552-1693) might represent the last….”“

  • This remark was accepted.

„Lines 376-377 The Authors write “In addition to these reliable results, the number of individuals in the breeds was sufficiently large.” I wonder what “sufficiently large” means in this context, is it in comparison to previous research or to some guidelines?“

  • This sentence was completed according to the guide.

This manuscript is a resubmission of an earlier submission. The following is a list of the peer review reports and author responses from that submission.

Round 1

Reviewer 1 Report

The paper is well written in style. However, I have following comments o the study

1. Use of SNP chip or whole genome sequencing data would have performed much better than the method used in this study. 

2. Data is dated (2015 and 2017)

3. Write up seems less professional and manuscript can be shortened at many places.

4.  They study is of local significance rather global implications. 

5.  Statistical testing needs reconsideration as there are lot of negative values. 

Reviewer 2 Report

The introduction is extremely long. It should be drastically reduced.

Whilst the paragraphs with the local breeds are very interesting, they should be shortened by 50%.

Also, many of the points regarding mitochondrial DNA work are well-known and should not be repeated.

Please indicate the locations where samplings were performed and also include a map of the country showing these locations.

All the primers and the details of the PCRs should be presented in tables inserted as supplementary material.

Please also include an analysis of the results in accord with the distance of sampling locations.

Table 1 should be moved in supplementary material

In the discussion, please compare the results with those about other European sheep breeds.

The manuscript also needs significant revision in English language, which is not good.